# Temporal Conformity-aware Hawkes Graph Network for Recommendations

## Abstract

Many existing recommender systems (RSs) assume user behavior is governed solely by their interests. However, the peer effect often influences individual decision-making, which leads to conformity behavior. Conventional solutions that eliminate indiscriminately such bias may cause RSs to neglect valuable information and depersonalize the recommendation results. Also, conformity can transform into user interest, e.g., discovering new tastes after a glance at popular music. By better representing different forms of conformity influence, we can do a better job at interest mining and debiasing. In certain extreme circumstances, the herd effect may be exacerbated by user anxiety with uncertainty (e.g., panic buying during the COVID-19 pandemic). RSs may thus fail to respond in time due to sudden and dramatic changes. Moreover, many existing studies potentially conflate conformity bias with popularity bias and lump together various factors responsible for differences in popularity. In this paper, we identify two distinct types of conformity behavior: informational conformity and normative conformity. To address this, we introduce the TCHN model, which utilizes attentional Hawkes processes to disentangle user self-interest and conformity in a personalized manner. Our approach incorporates temporal graph attention networks to capture users' stable and volatile dynamics. We conduct experiments on three real-world datasets, which uncover diverse levels of conformity among users. The results show that TCHN excels in recommendation accuracy, diversity, and fairness across various user groups [1].

## CCS Concepts

• **Information systems** → **Recommender systems**.

## Keywords

Interest, Conformity, Temporal Graph Attention Network, Recommendations

**ACM Reference Format:**
Anonymous Author(s). 2018. Temporal Conformity-aware Hawkes Graph Network for Recommendations. In *Proceedings of Make sure to enter the correct conference title from your rights confirmation email (Conference acronym 'XX)*. ACM, New York, NY, USA, 10 pages. https://doi.org/XXXXXXX.XXXXXXX

[1]Link to source code will be added here.

## 1 Introduction

Recommender systems utilize user behavioral data to capture user profiles and operate under the assumption that an individual's actions are intrinsically motivated by their interests [51]. However, in an informational cascade [5], people can frequently base their decisions not only on their own information but also on the influence of their peers within the same social community [39, 55]. Moreover, in different situations, people will actively shift the importance of the two factors to alleviate cognitive dissonance in their decision-making process [2, 4, 45]. For instance, when choosing a music album, users are more likely to rely on their own sensory experience than the opinions of others [49]; However, during extreme events such as the COVID-19 pandemic, the Fear of Missing Out (FOMO) phenomenon [46] comes into play, and people's purchasing behavior can be significantly influenced by others (e.g., panic buying in COVID-19 pandemic) [38, 70]. Due to the aforementioned fundamental assumption, traditional RSs are known to introduce popularity and conformity biases [10, 73, 74], where the learning process fails to differentiate between a user's genuine interest and the biased behavior.

Despite numerous studies [6, 33, 48, 50, 63, 73, 74] dedicated to this issue, several challenges remain to be addressed. First, most of the existing research views conformity behavior negatively as an interfering factor in user profiling [6, 48, 50]. However, we argue that conformity behavior reflects the combination of individuals' cognition of their inner needs [1, 39–41] and their perception of external circumstances [4, 45, 55, 70]. Eliminating behavioral data indiscriminately can result in the loss of valuable information and eventually lead to poor recommendations. Thus, we argue that a personalized method is needed to handle people's inner interest and the conformity bias in the system for each individual. While it is widely agreed that conformity causes users to blindly focus on popular items [33, 63, 73, 74], we may be overlooking individual differences in users' attitudes toward such items based on their personalities. For example, some users, known as "deep divers" in this research, have an unshakable interest in niche topics less influenced by environmental changes or others' opinions. While strategies aimed at eliminating biases may benefit these "deep divers", they may have a detrimental effect on another group of people who tend to prefer popular items and follow trends, denoted as "surfers" here.

Second, conformity bias and popularity bias are not equivalent. As the former is hardly observed and measured explicitly, many existing studies, e.g., [6, 63, 73, 74], have resorted to using *popularity* to characterize conformity. However, we argue that conformity behavior is only a sufficient but not necessary condition for popularity. The latter has its own complicated causes, such as the high quality of a product [7]. Individuals conform to peers either due to a lack of relevant knowledge (informational conformity) or to avoid isolation (normative conformity) [14, 16, 30]. This aligns with the fundamental logic of RSs, which is to intervene only when users' interests and behaviors do not match.

Third, these studies treat conformity bias as a *static* factor interfering with the operation of RSs. They learn conformity behavior inflexibly and disregard its temporal sensitivity and volatility. A potential interchange exists between user interest and conformity behavior: spontaneous conformity may help users develop new interests, whereas their enthusiasm for one item may potentially fade over time when the external stimulus wears off [4, 70]. Therefore, it is crucial to consider the temporal dimension of the relationship and how it may evolve over time.

Finally, the intensity and scale of conformity matter due to the fragility of information cascades [5, 22]. For example, some extreme outlier events (e.g., pandemic-like events [38, 39]) can cause abrupt population-scale concept drift in cascades. It further aggravates popularity bias [39] and out-of-distribution (OOD) issues [25] in RSs. Moreover, the population-scale concept drift might lead individuals to rush to products that normally go unnoticed, e.g., hand sanitizer during COVID-19. Conventional RSs are often unable to adapt to the changes, resulting in failure to balance recommendation accuracy and diversity [38, 39].

To tackle these challenges, we propose a **T**emporal **C**onformity-aware **H**awkes **N**etwork (TCHN), which decouples user behavior patterns in RSs as a mutual excited Hawkes process combining user interest and conformity. It draws inspiration from both Maslow's hierarchy of needs [40, 41] and social identity theory [55], which posit that individuals' decisions are influenced by their own inner needs as well as the opinions of others in their social environment:

$$D = \theta I + (1 - \theta)C, \tag{1}$$

where $D$, $I$ and $C$ represent "decision", "interest" and "conformity" respectively; $\theta$ is used to balance the influence of the two. It is worth noting that the motivation behind user behavior is a complex and difficult-to-observe sociological issue. In this study, we have only focused on the principal factors related to recommendation behavior. We disentangle their representations at the scale of the user-item interaction sequence graph. The resulting sequential embeddings are fed into customized attention-based Hawkes processes for preference prediction. TCHN treats user interests as stable information and conformity as a volatile and time-limited signal. Both user interest and conformity guide how to make a new decision, but only long-lasting user interest is stored in user representation. It dissociates the influence of user conformity behavior and personalizes its effect in the latest user profiling.

To sum up, the contributions of our research are as follows:

- We propose a Hawkes process-based attention graph network to disentangle interest and conformity in user decision-making. It delicately utilizes user conformity behavior in a personalized way instead of simply eliminating the bias.
- We model two forms of conformity behavior dynamically so that the model can differentiate the two and discover deep user interest during the interaction.
- We conduct extensive experiments on three real-world datasets that contain diverse population scales of conformity behavior. The results demonstrate that our model benefits both "surfers" and "deep divers" regarding recommendation accuracy and diversity.

## 2 Related Work

### 2.1 Conformity in User Interactions

Conformity pertains to the conduct of individuals who adhere to group norms in the presence of a non-conforming group that does not share their beliefs [13]. It is intrinsically motivated by individuals trusting the wisdom of crowds to make better decisions even though it may run counter to their own beliefs [55].

In the context of RSs, this behavior is manifested as users opting for popular products instead of ones that genuinely pique their interest. Besides, since the intrinsic motivation of conformity cannot be directly observed, several studies have reformulated the issue as a *popularity bias*. Schnabel et al. [50] and Saito et al. [48] propose inverse propensity scoring (IPS) based methods to eliminate popularity bias and distill user interests. Bonner and Vasile [6] guide the biased training process using a small set of unbiased intervention data. Zhang et al. [73] believe that the popularity of items can be a valuable indicator for learning user preferences. However, their approach treats popularity uniformly across all users without considering individual differences in preference. Zheng et al. [74] perceive popularity as an observable outcome of people's conformity and employ this signal to disentangle the causes of user decisions with finer granularity, i.e., user interests and conformity, by using causal mechanisms [26, 28]. However, it is not possible to discern the underlying motivation solely from the observed data. We argue that "conformity" is not a necessity for "popularity", i.e., it is too arbitrary to assert that users choose popular items out of conformity.

Deutsch and Gerard [16] elaborate on the two forms of conformity in social networks: informational and normative conformity. The former drives individuals to conform when they genuinely believe that the group is more knowledgeable or has better judgment in a particular situation, while the latter occurs when individuals conform to a group's beliefs or behaviors to gain social approval, avoid rejection, or fit in with the group [14]. Although conformity can sometimes be solely based on information or norms, it usually involves both factors simultaneously [30]. We believe this actually coincides with the underlying logic of RSs. Therefore, it is essential to leverage conformity behavior gently and intervene only when there are deviations in interest and behavior.

### 2.2 Sequential Recommendations

Information is disseminated in users' interactions with RSs. The interaction does not occur in isolation; it is an aggregation of the user's own needs and the influence exerted by preceding behaviors of peers [5]. While conformity is often seen as antithetical to user interests, it may turn into user interest in certain circumstances. Thus, we approach this issue in the framework of sequential recommendations. Sequential RSs model user temporal dynamics from their historical interactions. More recently, recurrent neural networks (RNNs) have become popular for sequential recommendations [17, 54, 75]. GRUs are typical implementations of RNNs to model user interactions by maintaining a memory of past inputs. Zhou et al. [75] proposes a two-layer GRU-based network to extract and model the evolution of users' deep interests. They innovatively introduce auxiliary loss to augment samples to alleviate the data sparsity problem. Transformer models are another prevalent class

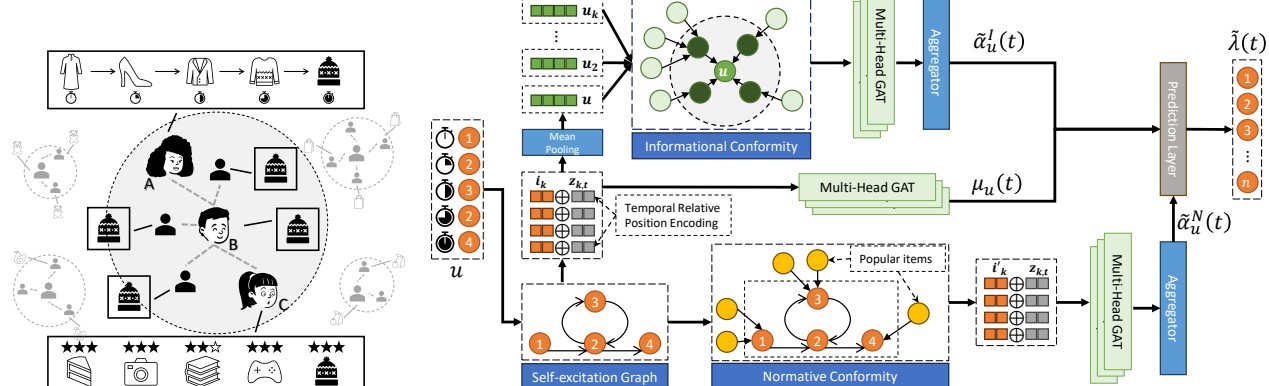

**(a) Diverse Conformity Types**                    **(b) TCHN Structure: Self-excitation, Informational Conformity, Normative Conformity**

of sequential neural network models that have achieved state-of-the-art performance in recommendation tasks [11, 54]. Attention mechanisms [57] can allow models to focus on specific parts of the input depending on the attributes of targets. Ying et al. [69] propose a hierarchical attention network for next-item recommendation tasks that balances user long-term preference evolution and new characteristics in their short-term preference.

## 2.3 Hawkes Process

Conventional sequential models can effectively capture ordinal dependencies between user behaviors but potentially neglect the time interval between two successive actions [35]. Temporal point processes (TPP) [15, 18] are mathematical abstractions of stochastic counting processes for realizations of asynchronous events that occur in continuous time. A TPP is denoted by $N(t)$, which is the number of events before time $t$. TPP has been widely applied in seismology [44], crime analysis [43] and user modeling [76] due to its predictive and explanatory abilities. Hawkes process (HP), a self-exciting variant of TPP, proposed by [24], assumes the arrival of an event stimulates subsequent events in the near future with various intensities. The mutual excitation property of HP facilitates user influence inference in social interactions. Recently, the misspecification in HP models has caused concern due to their pre-assumed intensity functions that do not fit the law of events [68]. Many researchers attempt to adapt sequential neural networks to TPP learning. The work in [42] extends HP by a long short-term memory (LSTM) network that relaxes the excitation constraint so as to simulate elevated or inhibited offspring events. Zhang et al. [72] and Zuo et al. [77] simultaneously and independently propose attention-based HPs which further extend classical HPs to capture deep timing dependencies between events and alleviate the gradient explosion and vanishing issues in RNNs. Besides, many studies devote to addressing various problems in RSs based on these fundamental models, e.g., preference evolution [3], repeat consumption modeling [59], adaptability in user long/short-term preferences [60].

## 2.4 Graph Representation Learning

To study the mutual excitation of user interaction events in Hawkes processes, it is essential to understand the topological structure of information diffusion in interactions. RSs, especially collaborative filtering (CF) models, often exhibit properties of social networks. This allows us to effortlessly convert user interactions into a graph structure. Leveraging recently popularized graph neural network (GNN) techniques in sequential RSs, we can learn the deep representations of user nodes in the social graph and understand the influence of neighboring nodes on their decision-making in information cascades. Li et al. [36] introduces a gated graph sequence neural network (GGNN) that can proficiently capture information propagation on directed graphs utilizing GRU components to filter pertinent information from neighboring nodes. However, the *permutation-invariant* aggregation function used by GGNN and its subsequent models in this paradigm [9, 62] during message passing can potentially result in the loss of valuable ordering information within the neighborhood [12]. Since the graph attention network (GAT) [58] was proposed, many applications in RSs have embraced GAT as a solution for addressing sequential graph recommendation problems (e.g., [61, 64]). In a GAT, attention mechanisms play a crucial role in determining the significance of each neighboring node with respect to a specific target node [58]. This allows the network to selectively attend to relevant neighbors and aggregate information from them while ignoring irrelevant nodes.

## 3 Methodology

Prior to elaborating on our approach, we aim to clarify the problem through the visual aid of Fig. 1a. The figure encapsulates three distinct user archetypes interacting with the hat, each driven by unique motivations. User $A$, characterized as a "deep diver", exhibits a self-motivated interaction with the hat (i.e., self-excitation); In contrast, user $B$ consults information from neighboring nodes with shared interests, culminating in the selection of the hat (i.e., informational conformity). Meanwhile, user $C$, typified as a "surfer", demonstrates a propensity for emulating peers and pursuing trending items (i.e., normative conformity). Our research endeavor is dedicated to a solution capable of discerning and accommodating

these heterogeneous motivations to provide adaptive recommendations.

## 3.1 Problem Formulation

Given a user set $\mathcal{U}$ ($M = |\mathcal{U}|$) and an item set $\mathcal{I}$ ($N = |\mathcal{I}|$), let $S_u(t) = [s_1, s_2, \cdots, s_k]$, with $s_k = (i_t, r_{i,t}, t)$, be a historical sequence of length $k$ for user $u$ and her recent $k$ engagements with the item set $\mathcal{I}$ up to time $t$, where $r_{i,t}$ is the feedback[2] of item $i$ at time $t$ given by user $u$. To elaborate further, we augment available data by reframing the historical sequence into a sequence matrix. This matrix includes supplementary paired samples structured as $\left[\left\langle S_u(t), i_{t+1}^+, i_{t+1}^- \right\rangle\right]_{t=1}^{k-1}$. Here, $i_{t+1}^+$ is the next item that user $u$ interacts with after $S_u(t)$ while $i_{t+1}^-$ is a hypothetical negative or unobserved instance for the same user. This allows us to capture the sequential nature of user behavior and model the evolution of user preferences over time [56].

Our final goal is to recommend a set of items for user $u$ that strike a balance between accuracy and diversity. Here, we present a Hawkes process based graph attention network TCHN (as shown in Fig. 1b) to address the problem of temporal evolution in user interests and conformity, which includes the following four main components: 1) **self-excitation** component to extract deep interest from users' own interaction history; 2) **informational conformity** modeling to aggregate influence of users' neighboring branches in an information cascade; 3) **normative conformity** modeling to measure the acceptance level of individuals on popular fashion; 4) **recommendation generation** component to fuse the three information signals to produce the final personalized recommendations. Before diving into the details of the model, we first introduce the preliminaries and the temporal relative position encoding method.

## 3.2 Preliminaries

*3.2.1 Hawkes Process* From a point process perspective, we treat the interaction ($s_t$) given by a user on an item as an *event* and the sequence involving the same user ($S_u(t)$) as a realization of a Hawkes process. Formally, in an infinitesimal time window $[t, t + \Delta t)$, the conditional intensity function $\lambda^*(t)$ [68] is:

$$\lambda^*(t) = \lambda(t \mid S_u(t)) = \lim_{\Delta t \to 0} \frac{\mathbb{E}[\Delta N(t) \mid S_u(t)]}{\Delta t}, \quad (2)$$

where $\mathbb{E}[\Delta N(t) \mid S_u(t)]$ is the expected number of events occurred in $(t+\Delta t]$ conditional on $S_u(t)$. Also, we assume two events coincide with probability 0, i.e., $\Delta N(t) \in \{0, 1\}$ [47]. The intensity $\lambda^*(t)$ denotes the occurrence rate of future events.

In this paper, we season the intensity function with the mutual excitation property of the Hawkes process. Also, we define an $M$-dimensional Hawkes process for all user interaction sequences where a single event in one dimension can excite or inhibit the intensities of all dimensions. The intensity function of $u$-th dimension can be updated as:

$$\lambda_u(t) = \mu_u(t) + \sum_{v \in U} \sum_{t_{v,i} < t} \alpha_{u,v}(t) \kappa_{u,v}(t - t_{v,i}), \quad (3)$$

where $\mu_u(t)$ is the base intensity of user $u$ at time $t$ triggered by their own intent. For the process, this arrival is spontaneous and

2w.l.o.g, the feedback can be explicit (e.g., ratings) or implicit (e.g., click).

independent of preceding events (a.k.a exogenous intensity). $t_{v,i}$ is the time when user $v$ interacted with item $i$. $\alpha_{u,v}(t)$ describes the strength of influence that user $v$ exerts on user $u$. Note that $\alpha_{u,v}(t)$ is unidirectional and not symmetrical, i.e., $\alpha_{u,v}(t) \neq \alpha_{v,u}(t)$. $\kappa_{u,v}(t - t_{v,i})$ is the triggering kernel that quantifies the rate of occurrence after the realization $t_{v,i}$. The most commonly used form of this kernel is the exponential kernel function: $\kappa_{u,v}(\Delta t) = e^{-\beta_{u,v}(\Delta t)}$, where $\beta_{u,v}$ controls the decay rate. The second addend in Eq. (3) captures the aggregated influence of other point processes (a.k.a endogenous intensity). A typical HP [24] only supports positive excitation, but here we relax such constraint and enable the inhibition effect [42, 60, 77], i.e., $\alpha_{u,v}(t), \mu_u(t) \in \mathbb{R}$.

In the context of RSs, the intensity function $\lambda_u(t)$ neatly crystallizes the user's decision-making process in Eq. (1). Users' intrinsic interest activates their base intensity; meanwhile, preceding events (interactions) from other individuals excite (or inhibit) the target user as a reference signal.

## 3.3 Embedding Representation

In our proposed model, we define three key entities within the latent embedding space: users, items, and time. To begin, the model initially maps one-hot representations of items to a unified low-dimensional embedding space, denoted as $i \in \mathcal{I} \mapsto \mathbf{i} \in \mathbb{R}^d$, where $d$ is the embedding dimension, and it is constrained such that $d \ll \min\{M, N\}$. Further, given two timestamps $\langle t_a, t_b \rangle$, we translate their time difference into a temporal relative positional embedding, i.e., $|t_b - t_a| \mapsto \mathbf{z}_{a,b}$. Consequently, the interaction sequence $S_u(t)$ can be encoded as latent vectors $S_u(t) \mapsto \mathbf{S}_u(t) = \left[\Theta\left(\mathbf{i}_\tau, \mathbf{z}_{\tau,t}\right)\right]_{\tau=1}^t$, where $\Theta(\cdot)$ is a differentiable, permutation-invariant function, such as element-wise summation or concatenation.

Remarkably, we do not explicitly encode user profiles, but instead aggregate their historical interaction sequences $S_u(t)$ to serve as user representations. Formally, user $u$ is mapped to the embedding $\mathbf{u} := \Phi(S_u(t))$, where $\Phi(\cdot)$ is a differentiable function. The implementation details of $\Theta(\cdot)$ and $\Phi(\cdot)$ will be discussed in subsequent sections.

## 3.4 Temporal Relative Position Encoding

As outlined in Sec. 3.2.1, our approach involves scrutinizing the user's interaction history within the Hawkes point process framework. This methodology entails observing how new interactive events can either excite or inhibit the intensity of ongoing events, with these effects gradually diminishing over time according to the specified kernel function $\kappa_{u,v}(\Delta t)$ in Eq. (3). In contrast to RNN networks, standard attention-based transformer models lack the capability to discern position differences in the input sequence [52, 57]. Consequently, our learning model necessitates the inclusion of supplementary positional representations within the input data. Traditional order-position encoding [57] or absolute value of time cannot disclose rigorous time transition information as effectively as relative timespan [67]. In this work, we revamp the approach proposed by [52, 67] into our model to expose the pairwise temporal relationships between different interactions. Formally, we first establish a positive semi-definite (PSD) temporal kernel $\kappa(t_a, t_b)$ that complies with Bochner's theorem assumption (proof

is supplied in Appx. A). Then, the kernel value is quantized to an integer, serving as the relative position for events at time $t_a$ and $t_b$:

$$\kappa(t_a, t_b) = \theta_z \ln\left(\beta_{a,b}\left(|t_b - t_a|\right) + 1\right) \tag{4}$$

$$z_{a,b} = \lceil \kappa(t_a, t_b) \rceil, \tag{5}$$

where $\theta_z > 0$ is a hyper-parameter to manage the granularity and capacity for the positional embedding space. $\beta_{a,b} > 0$ governs the decay rate for different events. We stipulate that $\theta_z = 10$ and the time unit of $t_a$ and $t_b$ are *days* in our experiment settings. Additionally, we postulate that different individuals share the same sensitivity to time differences for simplicity; hence, we set $\beta_{a,b} = \beta = 1$. This assumption implies that recent events exert a more subtle influence on their immediate decisions, while events more distant from the present have a diminishing and less pronounced impact. Finally, we encode the relative positional information into a position-embedding matrix, where $z_{a,b} \in \mathbb{R}^+ \mapsto \mathbf{z}_{a,b} \in \mathbb{R}^d$.

In the following sections, we will provide detailed insights into the various components of our methodology as integrated within the context of the Hawkes process and a graph neural network framework.

## 3.5 Self-excitation Graph

Intuitively, the base intensity $\mu_u(t)$ of a user on a specific item at time $t$ is primarily determined by their intrinsic interest. In addition, individuals' interests are subject to change over time as their inner needs evolve [31, 39]. We derive nutritional value from the user's historical interaction sequence. Additionally, more recent interactions carry greater weight in determining the user's current interests and needs.

To implement this concept, we transform each interaction sequence $S_u(t)$ into a user-specific graph. This enables the acquisition of item-transition patterns within the sequence $S_u(t)$ by utilizing a time-aware graph attention network (TGAT). Given $S_u(t)$, let $G_u(t) = (V_u(t), E_u(t))$ represent a directed user-specific graph, where $V_u(t) \subset \mathcal{I}$ denotes the set of interacted items in $S_u(t)$, and $E_u(t)$ is the edge set. Each edge connects two adjacent items $(i_a, i_b)$ in $S_u(t)$ and points from $i_a$ to $i_b$ if $t_b > t_a$. Next we introduce TGAT.

### 3.5.1 Time-aware Graph Attention Layer

Analogous to GAT [58], the TGAT layer can be conceived as a local aggregation operator. It takes as input the hidden representations and temporal relative position embeddings of the neighborhood surrounding the target node. The output is the time-aware representation for the target node at the current time $t$.

As mentioned in Sec. 3.3, given current timestamp $t$, the hidden representation of $G_u(t)$ is specified by

$$\mathbf{H}_u(t) = \Theta\left(\mathbf{I}_u(t), \mathbf{Z}_u(t)\right)^\top = [\mathbf{i}_{t_1} + \mathbf{z}_{t_1, t}, \cdots, \mathbf{i}_{t_k} + \mathbf{z}_{t_k, t}]^\top \tag{6}$$

where $\mathbf{I}_u(t) = [\mathbf{i}_{t_k}]_{i \in G_u(t)}$ is the node embedding matrix of $G_u(t)$, and $\mathbf{Z}_u(t) = [\mathbf{z}_{t_k, t}]_{t_k \in G_u(t)}$ is the temporal embedding matrix of relative positions between nodes in $G_u(t)$. $\Theta(\cdot)$ is chosen as element-wise summation.

Subsequently, we pass $\mathbf{H}_u(t)$ and the embedding of the target item $\mathbf{i}_t$ into the self-attention module. And the attention output

will be treated as the base intensity of user $u$ on item $i$:

$$\tilde{\mu}_{u,i}(t) = Attention(\mathbf{H}_u(t), \mathbf{i}_t) = \text{softmax}\left(\frac{\mathbf{H}_u(t)\mathbf{i}_t^\top}{\sqrt{d}}\right) \tag{7}$$

Finally, Eq. (7) models the evolution of the interest of user $u$, and captures her intrinsic preference on the target item $i$.

## 3.6 Conformity Graphs

The classical Hawkes process inherently captures the strength of unidirectional influence from the predecessors to the target user, i.e., $\alpha_{u,v}(t)$ in Eq. (3). However, we speculate that this influence entangles the conformity of individuals. In other words, interactions between individuals are impacted not only by their shared preferences but also by peer effects. We extend the classical model to disentangle various forms of conformity, namely informational conformity and normative conformity [16]. As aforementioned, the two forms of conformity often occur concurrently [30], and their influences usually vary given different users and items. Therefore, inspired by [34], we disentangle the influence strength $\alpha_{u,v}(t)$ into two additive terms:

$$\alpha_{u,v}(t) = \theta_{u,v}^I(t)\alpha_{u,v}^I(t) + \theta_{u,v}^N(t)\alpha_{u,v}^N(t), \tag{8}$$

where $\alpha_{u,v}^I(t)$ and $\alpha_{u,v}^N(t)$ are the two forms of conformity respectively. The time-varying coefficients $\theta_{u,v}^I(t)$ and $\theta_{u,v}^N(t)$ balance their contributions at time $t$. The intensity function of the *conformity-aware Hawkes process* can be updated as follows by substituting it to Eq. (3):

$$\lambda_u(t) = \mu_u(t) + \sum_{v \in U} \sum_{t_{v,i} < t} \left(\theta_{u,v}^I(t)\alpha_{u,v}^I(t) + \theta_{u,v}^N(t)\alpha_{u,v}^N(t)\right) \kappa_{u,v}(t - t_{v,i}) \tag{9}$$

Next, we describe how to quantify the two forms of conformity in RSs. Before that, we need to establish a clear scope and definition for both within the framework of RSs.

### 3.6.1 Informational Conformity

In the context of RSs, informational conformity could manifest itself when users rely on the recommendations of the system because they believe that the system has more information and knowledge about the items being recommended. Formally, we define **informational conformity** as the extent to which users adopt the aggregated preferences of other *like-minded users* (neighbors) in the system [51].

We formulate the modeling of informational conformity in a GNN, allowing for effective learning of mutual influences between nodes. Given the user set $\mathcal{U}$, we build a conformity graph based on the social connections between users. Without loss of generality, in the framework of RSs, we infer the social connection between users through the interaction between users and items instead of introducing additional social information, such as social actions (e.g., following and liking) [64]. This is because merely understanding the structure of a social action-based topology is inadequate to tackle this issue, as the connections between individuals do not always imply the occurrence of social interactions among them, whereas they can be affected by some strangers unintentionally [34]. For example, suppose both users $u$ and $v$ interacted with the same item $i$ at time $t_u$ and $t_v$. We build an edge pointing from $u$ to $v$ if $t_v > t_u$ that indicates user $u$'s action can potentially affect

(excite or inhibit) user $v$'s decision even though they do not know each other. Thus the social network is formulated as a user-user directed graph: $G = \langle \mathcal{U}, E \rangle$, where $E$ is the set of social connections among $\mathcal{U}$. Particularly, the weight of the edge between $u$ and $v$ indicates the social influence of user $u$ on user $v$. According to social psychology theories [2, 16, 71], the degree of influence exerted by user $u$ positively correlates with the level of informational conformity exhibited by user $v$ towards user $u$. Based on this, it can be inferred that if individuals $v$ and $u$ interact *frequently*, it is advisable to enhance their informational influence. Additionally, we hypothesize that user $v$ will likely conform to user $u$ when user $v$ consistently agrees with user $u$ (*opinion polarity* w.r.t an item). Thus, given a historical interaction sequence $S_u(t)$ (resp. $S_v(t)$) of user $u$ (resp. $v$) the weight of edge $u \rightarrow v$ can be quantified as:

$$w_{u,v} = \frac{\sum_{(r,t) \in \Lambda(u,v)} (r_u - \bar{r}_u)(r_v - \bar{r}_v) \exp\left(-\beta_{u,v}(t_v - t_u)\right)}{\sqrt{\sum (r_u - \bar{r}_u)^2 \sum (r_v - \bar{r}_v)^2}} \quad (10)$$

where $\Lambda(u,v) = \{S_u(t) \cap S_v(t) \mid i_u = i_v \wedge t_v >_u\}$ is the intersection of ratings of items interacted by both $u$ and $v$; meanwhile, $t_v > t_u$ in each interaction. Besides, in line with the definition of the Hawkes process in Eq. (3), we penalize the opinion polarity far from the present by the decay function. That is, the weight reveals how likely it is that the decision of user $v$ is infected by user $u$ in the presence of informational conformity.

We apply a GAT on the social influence graph to learn about the informational conformity strength in the ego social network of the target user. The GAT layer utilizes a multi-head masked attention mechanism to aggregate the influence of neighboring nodes on the target node. Firstly, the attention coefficient between the target node $v$ and one of its neighboring nodes $u$ at the $k$-th head is:

$$\mathbf{h}_u = \Phi\left(\mathbf{H}_u(t)\right) = \frac{1}{k} \sum \left(\mathbf{i}_{t_k} + \mathbf{z}_{t_k}\right), \mathbf{h}_v = \Phi\left(\mathbf{H}_v(t)\right) \quad (11)$$

$$c_{u,v}^k = \phi\left(\left[w_{u,v}(\mathbf{W}_k \cdot \mathbf{h}_u) || (\mathbf{W}_k \cdot \mathbf{h}_v)\right]\right), \quad (12)$$

$$\alpha_{u,v}^k = Attention\left(c_{u,v}^k\right) = \text{softmax}\left(\frac{c_{u,v}^k}{\sqrt{d}}\right), \quad (13)$$

where $\phi(\cdot)$ is the LeakyReLU activation function, $w_{u,v}$ is the weight of edge $u \rightarrow v$, $\mathbf{W}_k$ is the shared parameter matrix and $\mathbf{h}_u$ (resp. $\mathbf{h}_v$)) is the node embedding of $u$ and $v$, which is aggregated from self-excitation graph $G_u(t)$; $[\cdot||\cdot]$ is the concatenation operation of embeddings.

Its node representation is computed by aggregating the influence from the neighboring nodes:

$$\mathbf{h}_v'(K) = \frac{1}{K} \sum_{k=1}^{K} \sigma\left(\sum_{u \in \mathcal{N}_v} \alpha_{u,v}^k \mathbf{W}_k \mathbf{h}_u\right), \quad (14)$$

where $\sigma(\cdot)$ is the Sigmoid activation function. We apply a *mean-pooling* operation to fuse the $K$-head output. The informational influence strength will be:

$$\tilde{\alpha}_{u,v}^I(t) = \frac{1}{K} \sum_{k=1}^{K} \alpha_{u,v}^k. \quad (15)$$

### 3.6.2 Normative Conformity
Normative conformity refers to the phenomenon in which individuals unconsciously conform to the norms of a particular social group, suppressing their rational thinking [55]. In the context of RSs, this is evident when users change their usual behavior habits and preferences to align with those of the crowd (e.g., panic buying [70]), or adjust their own evaluations to adhere to the average opinion [20]. Formally, we define **normative conformity** as the extent to which users adopt the aggregated preference on *recently popular items* within the system. Specifically, we speculate that more popular items and recent interactions can convey stronger conformity signals to the target user. Then, the target user can respond to the signals by shifting their original intentions. For instance, online shopping users refer to the sale promotion on the home page (e.g., "Best sellers" in Amazon [51]) to place an order; music platform users add recent hits to their playlists [49].

Following this definition, to quantify the attractiveness of fashion to a user during the decision-making process, we gather the most popular items $P(t)$ in the system within the recent time window. The width of the time window is denoted as $\tau$, meaning that the interactions considered must have occurred at a time $t$ such that $t \geq (t_d - \tau)$. Here, $t_d$ represents the time at which the user makes a decision or receives a recommendation.

We first map the items in the collection $P(t)$ to their embedding space: $P(t) \in \mathbb{R}^n \mapsto \mathbf{P}(t) \in \mathbb{R}^{n \times d}$, where $n = |P(t)|$. Next, we compute the popularity-weighted attention score, which can be treated as the normative conformity strength:

$$c\left(\mathbf{P}(t), i\right) = \mathop{||}_{j=1}^{n} \phi\left(\frac{p_j}{p_i} \cdot \mathbf{W} \cdot \mathbf{j}\right), \quad (16)$$

$$\alpha\left(c\left(\mathbf{P}(t), i\right)\right) = \text{softmax}\left(\frac{c\left(\mathbf{P}(t), i\right) \mathbf{i}^\top}{\sqrt{d}}\right), \quad (17)$$

where $\mathop{||}_{j=1}^{n} (\cdot)$ is $n$ concatenation operations, $\phi$ is the ReLu activation function. Importantly, we inject a relative popularity weight to the attention coefficient $c\left(\mathbf{P}(t), i\right)$, i.e., $p_j/p_i$, where $p_j$ (resp. $p_i$) is the number of interactions of item $j$ (resp. $i$) within the time window $\tau$. Specifically, the attention score amplifies the impact of more popular items while suppressing the influence of less popular items, which will be treated as the normative influence strength in our model: $\tilde{\alpha}_i^N(t) = \alpha\left(c\left(\mathbf{P}(t), i\right)\right)$.

## 3.7 Recommendation Generation

Through the analysis of the above models and sociological research [2, 13, 71], we believe that the two forms of conformity play different roles in shaping individuals' beliefs and behavior patterns. Specifically, informational conformity has the potential to induce genuine and enduring alterations in beliefs. The result of informational influence is normally private acceptance: the development of the interest of individuals. Instead, normative conformity is less likely to cause lasting change and more of a coping mechanism people use to avoid isolation. Once the external stimulus diminishes, individuals consciously correct this behavior. However, prior to that, this conformity signal plays an important role in influencing people's decision-making.

**Table 1: Statistics of Datasets**

| Dataset | #Users | #Items | #Interactions |
|---------|--------|--------|---------------|
| Diginetica | 72,013 | 29,453 | 580,490 |
| Kuai | 7,176 | 10,612 | 1,153,787 |
| Yelp | 78,163 | 57,718 | 1,856,942 |

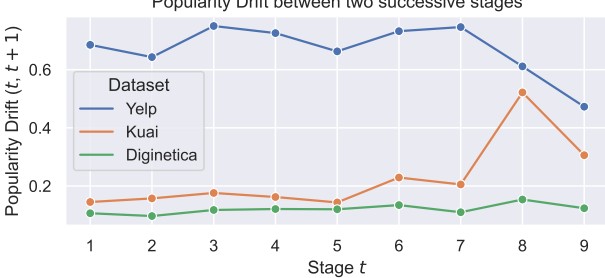

**Figure 2: Popularity Drift between $t$ and $t + 1$**

Therefore, when training the model, we only update the representations of users and items in *self-excitation* and *informational conformity* components. *Normative conformity* only participates in decision-making without affecting the deep representations.

The preference of the target user $u$ on the target item $i$ at time $t$ is quantified as the intensity of the conformity-aware Hawkes process:

$$\tilde{\lambda}_{u,i}(t) = \tilde{\mu}_u(t) + \sum_{v \in U} \theta^I_{u,v}(t)\tilde{\alpha}^I_{u,v}(t) + \theta^N_{u,v}(t)\tilde{\alpha}^N_i(t). \quad (18)$$

L2 regularization is employed on the parameters of the mapping function to address overfitting, although it is not demonstrated here for brevity.

We apply a contrastive learning scheme to train the model. Given the sequence $\left[ < S_u(t), i^+_{t+1}, i^-_{t+1} > \right]_{t=1}^{k-1}$, the loss function can be computed as:

$$L = -\frac{1}{k} \sum_{t=1}^{k} \left( y \log \tilde{\lambda}_{u,i^+}(t) + (1 - y) \log(1 - \tilde{\lambda}_{u,i^-}(t)) \right) \quad (19)$$

## 4 Experiments

### 4.1 Experiment Settings

*4.1.1 Datasets* We evaluate the proposed model on three real-world datasets: Diginetica [3], Kuai [21] and Yelp [4], because: 1) their degrees of sparsity vary and are collected from different domains; 2) their users have different degrees of interest and conformity drift. We map the labels of all positive interactions to "1", and others to

---
[3] https://competitions.codalab.org/competitions/11161
[4] https://www.kaggle.com/datasets/yelp-dataset/yelp-dataset

"0". Following [65, 66], we extract a 5-core dense subset covering years 2018-2021 for Yelp datasets, including only users and items with more than 5 interactions. Its users potentially experienced drastic changes in the outlier environment [53]. To show such drift, we divide these datasets evenly into 10 stages in chronological order and measure the popularity drift between successive stages [73]. We firstly collect the relative popularity of items in each stage $[p^i_1, p^i_2, \cdots, p^i_t]$. Then we apply Jensen-Shannon Divergence (JSD) [19, 73] to measure the similarity between two distributions in different stages. The trends in Fig. 2 show Yelp contains a larger scale of drift while Diginetica and Kuai can be seen as normal situations. The statistics of these datasets are summarized in Table 1. We partitioned the datasets into three sets - training, validation, and testing - in chronological order using Leave-one-out for each user, applied to all baselines.

*4.1.2 Baselines* Nine baselines from four main categories are selected in our experiments, including: *Conventional*: MF [32]; *Debiased*: MF_REL [48], DICE [74], PDA [73]; *Sequential*: GRU4Rec [27], SASRec [29]; *Graph based*: LESSR [12], GCEGNN [62], NISER [23].

*4.1.3 Metrics* nDCG is applied to measure the accuracy of recommendations; and Mean Intra User Distance (MIUD) [8], Tail Percentage (Tail) [37] are used to measure *intra-user* and *inter-user* diversity. We collect top-$k$ = 2 and 10 recommendation results to calculate these metrics.

### 4.2 Performance Comparison

As shown in Table 2, the proposed TCHN outperforms all baselines in both accuracy and diversity from multiple perspectives.

*4.2.1 Performance in Normal Situations* The results on Diginetica and Kuai datasets demonstrate the advantage of sequential and GNN-based models over debiased models. However, the proposed TCHN achieves great improvement when compared with baselines. This is because TCHN disentangles users' interest and conformity signals from their historical interactions, allowing them to play distinct roles in prediction based on their unique characteristics. As a result, TCHN can recommend niche items that align with users' interests rather than solely popular ones. It is particularly evident in short video datasets (Kuai), as social attributes are prevalent on these platforms, making users more susceptible to influence and potentially leading to the development of new interests and tastes.

*4.2.2 Performance in Unusual Situations* The Yelp (2018-2021) dataset effectively captures the population-scale shift in user behavior during the pandemic [53]. Conventional debiased methods falter in generating personalized recommendations in such context, due to outdated user preferences [39]. Specifically, all baseline methods encounter difficulties in offering diverse recommendations when users exhibit homogeneous behavioral patterns. However, our model sustains optimal diversity without compromising accuracy. It adeptly capitalizes on users' conformity behavior and aligns their previous interests to deliver effective personalized recommendations.

### 4.3 Ablation Studies

Given the four components demonstrated in Section 3, we conduct several ablation studies by insulating different components.

**Table 2: Performance on Three Datasets: Best results are highlighted in bold, while the second-best results are underlined.**

| Model | Normal Situations | | | | | | | | Unusual Situations (e.g., Pandemic) | | | |
| --- | --- | --- | --- | --- | --- | --- | --- | --- | --- | --- | --- | --- |
| | Diginetica | | | | Kuai | | | | Yelp | | | |
| | nDCG@2 | nDCG@10 | MIUD@10 | Tail@10 | nDCG@2 | nDCG@10 | MIUD@10 | Tail@10 | nDCG@2 | nDCG@10 | MIUD@10 | Tail@10 |
| MF | 0.0574 | 0.1730 | 0.583 | 0.0051 | 0.1856 | 0.2102 | 0.660 | 0.0059 | 0.0041 | 0.0076 | 0.601 | 0.0045 |
| MF_REL | 0.0659 | 0.1785 | 0.578 | 0.0058 | 0.1904 | 0.2212 | 0.656 | 0.0065 | 0.0045 | 0.0085 | 0.703 | 0.0049 |
| DICE | 0.0984 | 0.1989 | 0.588 | 0.0089 | 0.2168 | 0.2503 | 0.655 | 0.0099 | 0.0062 | 0.0101 | 0.703 | 0.0054 |
| PDA | 0.1172 | 0.2004 | 0.566 | 0.0095 | 0.2203 | 0.2565 | 0.623 | 0.0101 | 0.0058 | 0.0095 | 0.699 | 0.0056 |
| GRU4Rec | 0.1271 | 0.2142 | 0.604 | 0.0154 | 0.2545 | 0.2850 | 0.697 | 0.0250 | 0.0134 | 0.0278 | 0.693 | 0.0171 |
| SASRec | 0.1406 | 0.2312 | 0.604 | 0.0166 | 0.2587 | 0.2878 | 0.693 | 0.0280 | 0.0158 | 0.0332 | 0.697 | 0.0174 |
| LESSR | 0.1391 | 0.2297 | 0.603 | 0.0143 | 0.2618 | 0.2907 | 0.696 | 0.0276 | 0.0154 | 0.0303 | 0.688 | 0.0079 |
| NISER | 0.1506 | 0.2432 | 0.603 | 0.0131 | 0.2576 | 0.2862 | 0.696 | 0.0274 | 0.0153 | 0.0314 | 0.688 | 0.0077 |
| GCEGNN | 0.1555 | 0.2489 | 0.603 | 0.0178 | 0.2569 | 0.2891 | 0.696 | 0.0268 | 0.0177 | 0.0350 | 0.688 | 0.0174 |
| THN-si | 0.1546 | 0.2452 | 0.603 | 0.0171 | 0.2646 | 0.2902 | 0.696 | 0.0424 | 0.0183 | 0.0354 | 0.658 | 0.0077 |
| THN-sn | 0.1323 | 0.2132 | 0.611 | 0.0146 | 0.2577 | 0.2871 | 0.706 | 0.0373 | 0.0144 | 0.0328 | 0.698 | 0.0177 |
| TCHN | **0.1615** | **0.2510** | **0.624** | **0.0196** | **0.2686** | **0.2989** | **0.703** | **0.0476** | **0.0197** | **0.0366** | **0.724** | **0.0182** |
| *p*-value | <0.001 | <0.001 | <0.001 | <0.001 | <0.001 | <0.001 | <0.001 | <0.001 | <0.001 | <0.01 | <0.001 | <0.001 |

**Table 3: Comparison of position encoding methods**

| | Diginetica | Kuai | Yelp |
| --- | --- | --- | --- |
| | nDCG@10 | nDCG@10 | nDCG@10 |
| TRP | **0.2510** | **0.2989** | **0.0366** |
| OP | 0.2340 | 0.2901 | 0.0345 |
| RP | 0.2485 | 0.2920 | 0.0350 |

*4.3.1 Impact Temporal Relative Position Encoding* Inspired by the Hawkes process, we introduce the temporal relative position (TRP) encoding method, which enhances the temporal awareness capabilities of traditional attention mechanisms. To validate the effectiveness of this approach, we substitute our position encoding with other commonly employed techniques such as order-position (OP) encoding in [29] and reversed position (RP) encoding in [62]. Table 3 illustrates the performance of various methods across three datasets. Clearly, the temporal relative position encoding surpasses other comparative settings, as it adeptly captures the temporal dependencies within interaction sequences.

*4.3.2 Conformity Modeling* This section demonstrates the effect of *informational conformity* and *normative conformity* components. We eliminate the normative conformity layer, denoted as "THN-si", and the informational conformity layer, denoted as "THN-sn" respectively. As shown in Table 2, "THN-si" can still outperform other baselines w.r.t accuracy and diversity on Diginetica and Kuai datasets, which reveals the feasibility and effectiveness of integrating the informational conformity model into TCHN. However, it is unable to detect irrational user behaviors in interactions due to similar limitations as conventional RS models. As a result, it loses too much diversity score in the Yelp dataset, potentially misinterpreting normative conformity behavior as users' genuine interest. On the other hand, "THN-sn" fails to generate comparable recommendations as baselines due to its failure to acknowledge the positive impact of like-minded neighbors on the user in question. In "THN-sn", users cannot get nourishment and develop new interests from those who they think are knowledgeable predecessors. In summary, TCHN can significantly improve upon incomplete models, particularly in short-video data and unusual situations.

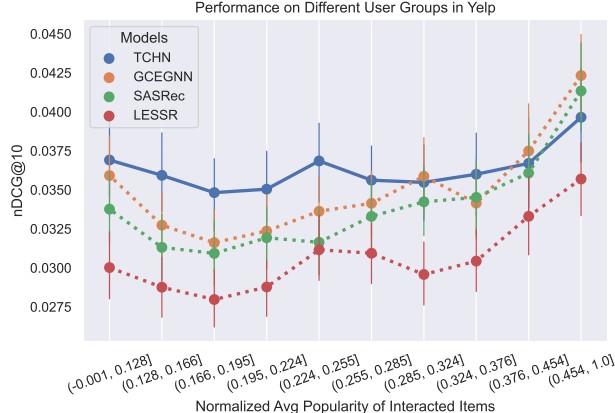

**Figure 3: Performance on Different User Groups in Yelp**

## 4.4 Studies on Different User Groups

As aforementioned, we define two groups of users: "surfers" and "deep divers" based on their perception of conformity and taste for popular items. As shown in Fig. 3, we evaluate our model on different user groups to confirm that the recommended items can benefit all users with various perceptions of conformity and taste for popular items. Here, we divided users into ten groups according to the normalized average popularity of the items they interacted with. For each group, we measure the *n*DCG@10 of the recommendations. We can see that TCHN achieves consistent performance across all groups of users, while other baselines clearly favor "surfers" who like popular items.

## 5 Conclusion

Aiming at the problem that conventional RSs potentially ignore the positive effect of users' conformity behavior, we propose a TCHN for disentangling user interest and conformity based on attentional Hawkes process networks. This paper investigates how to model the user interest and the conformity evolution at the individual level to maximize their effect for recommendation purposes, and the experiment results show that modeling in this way leads to better recommendations in terms of accuracy and diversity.

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

# A Appendix: Proof of Positive Definiteness of the Kernel Function

In this appendix, we provide a proof that the kernel function $\kappa = \theta_z \ln \left( \beta_{a,b} \left( |t_b - t_a| \right) + 1 \right)$ is positive definite, given $\theta > 0$ and $\beta > 0$, thus satisfying the assumption of Bochner's theorem.

First, let us recall the definition of Bochner's theorem:

*Definition A.1 (Bochner's Theorem).* A complex-valued function $f$ on a locally compact abelian group $G$ is positive definite if and only if $f$ is the Fourier transform of a positive measure on the dual group $\hat{G}$.

PROOF. Let $N \in \mathbb{N}$ and $t_1, t_2, \ldots, t_N \in \mathbb{R}^+$ be arbitrary. We must show that the matrix $[\kappa(t_i - t_j)]_{i,j=1}^N$ is positive semi-definite. This means for any vector $x = (x_1, x_2, \ldots, x_N) \in \mathbb{R}^N$, we have

$$\sum_{i=1}^N \sum_{j=1}^N x_i \overline{x_j} \kappa(t_i - t_j) \geq 0.$$

Substituting the expression for $\kappa$, we get

$$\sum_{i=1}^N \sum_{j=1}^N x_i \overline{x_j} \theta_z \ln \left( \beta_{a,b} \left( |t_i - t_j| \right) + 1 \right) \geq 0.$$

Since $\theta_z > 0$ and $\ln(x + 1)$ is increasing for $x \geq 0$, it suffices to show that $\sum_{i=1}^N \sum_{j=1}^N x_i \overline{x_j} \beta_{a,b} \left( |t_i - t_j| \right) \geq 0$.

This follows from the fact that the function $|t_i - t_j|$ is non-negative, and the product of non-negative numbers is non-negative. Hence, the kernel function $\kappa = \theta_z \ln \left( \beta_{a,b} \left( |t_b - t_a| \right) + 1 \right)$ is positive definite. □

This confirms that our kernel function satisfies the assumptions of Bochner's theorem, and can therefore be used in the context of harmonic analysis and related mathematical fields.

