# OpenReview forum: "Temporal Conformity-aware Hawkes Graph Network for Recommendations"
_ACM.org/TheWebConf/2024/Conference — TheWebConf24 Oral_

### Official Review · Reviewer_J4xK · 2023-11-21

**Novelty:** 5
**Technical Quality:** 5

**Review:**

### Summary

This work studies on temporal conformity bias, and proposes a new method based on Hawkes graph neural network. This work has the following strengths and weaknesses:


### Strengths

1. This work studies on an important problem.
2. This work proposes a new debiasing method --- leveraging Hawkes process in modeling conformity effect is interesting and reasonable.
3. Extensive experiments are conducted to validate the effectiveness of the proposal.
4. This work is well-writing, and the proposed method is well-motivated.

### Weaknesses

1.	My major concern is the missing of some important related work. In fact, temporal conformity bias has been studied by [a1], which disentangles the popularity bias into static item quality and the temporal conformity effect. I understand this work employs different techniques from [a1]. The difference should be explicitly discussed in related work and [a1] should be considered as a baseline in the experiments.

[a1] TKDE’22: Popularity bias is not always evil: Disentangling benign and harmful bias for recommendation

Some minor ones:

2.	Note that a Hawkes process model is usually learned with a specific log-likelihood loss. However, this work simply employs a contrastive loss. The rationality should be discussed.

3.	It would be better to give some future directions, which may inspire more research work on this topic.

### Overall evaluation

In summary, while this work has some limitations, I appreciate this work studies an important problem and proposes a new method. Besides, this work is very well-writing. As such, I give accept.

**Questions:**

Please refer to weaknesses.

**Ethics Review Description:**

None.

**Reviewer Confidence:**

4: The reviewer is certain that the evaluation is correct and very familiar with the relevant literature

**Scope:**

4: The work is relevant to the Web and to the track, and is of broad interest to the community

---

### Official Review · Reviewer_spRy · 2023-11-24

**Novelty:** 6
**Technical Quality:** 7

**Review:**

In this paper, the authors deal with sequential recommendation. The recommendations are produced with graph neural networks that disentangle user inartistic preferences from their preferences caused by conformity (informational and normative). To this end, the graph neural network models Hawkes process, hence the "Temporal Conformity-aware Hawkes Graph Network".

The paper is very well written, it connects well the abstractions of the model to the theoretical background (conformity models). The state of the art is well covered, separating nicely literature on topics of interest. The evaluation procedure is done adequately, it covers the baselines, and significance analysis is as well done.

**Questions:**

What are the limitations of this work?
Why none of the baselines are non-neural approaches?

**Ethics Review Description:**

-

**Reviewer Confidence:**

3: The reviewer is confident but not certain that the evaluation is correct

**Scope:**

4: The work is relevant to the Web and to the track, and is of broad interest to the community

---

### Official Review · Reviewer_dJwi · 2023-11-26

**Novelty:** 4
**Technical Quality:** 4

**Review:**

The paper addresses the problem of incorporating conformity into a recommendation system, while considering temporal modeling via a Hawkes process. I am outsider to this area and I am not able to judge well the technical novelty and originality of the work, nor the technical correctness. As an outsider, however, the problem statement and methodology look reasonable to me, and the empirical evaluation presents positive results.

**Questions:**

No specific questions.

**Ethics Review Description:**

No ethics issues.

**Reviewer Confidence:**

1: The reviewer's evaluation is an educated guess

**Scope:**

3: The work is somewhat relevant to the Web and to the track, and is of narrow interest to a sub-community

---

### Official Review · Reviewer_47gL · 2023-11-26

**Novelty:** 5
**Technical Quality:** 5

**Review:**

The authors propose an approach for disentangling personal preferences from influences from outside (conformity).

It is not clear if the work addresses individual- or group-RSs. Majority of peer-pressure work has been done in group RS. Related work is missing important works on group RS and conformity (e.g. Nguyen and RIcci,https://link.springer.com/article/10.1007/s11257-019-09240-9), and emotional contagion (Kramer, Adam D. I., Jamie E. Guillory, and Jeffrey T. Hancock. “Experimental Evidence of Massive-Scale Emotional Contagion through Social Networks.” Proceedings of the National Academy of Sciences 111, no. 24 (June 17, 2014): 8788–90. https://doi.org/10.1073/pnas.1320040111.
).

Second, the proposed model, which is quite complex, includes many choices that are not welll supported. For example, why, in Sect 3, are there the three user types (referred to in Fig 1a)?

What I miss in the datasets chosen and in the scenarios is the level of user interaction with others, which, in the studies so far, has demonstrated various degrees of peer pressure (in whatever form you like, from emotional contagion to conformity). The datasets were chosen to have diverse popularity drift. I don't think this is a valid reason.

In the beginning the authors claim that they aim at disentangling personal preferences from the ones that come from outside. However, the evaluation is a classical precision-based (NDCG). A RS that aims at disentangling reasons for behaviour should not e measured by precision. The goal of such a system is not to achieve higher precision.

**Questions:**

I am concerned about the extent of usage of LLMs in this paper. Along with three more papers I reviewed, they seem to be
done from a template rather than through a creative scientific effort. I am expressing my doubt that it (and the three others) is human-written.

They share some concerning similarities. They all have a similar structure: high-level problem description, non-scientific problems (very
industry-oriented, non generalizable), incoherent related work, weak motivation/positioning within related work, overcomplicated (for the
task at hand) methodology, IR-based metrics (even if the problem addressed calls for other types of metrics), questionable baselines (for which it's not clear whether they have been optimized as the proposed approach), ablation study. The papers are also similar visually (figures, tables). It almost seems like the methodology is so complicated in order to obfuscate.

I hope I am wrong. I would appreciate if the  authors could address my concerns.

**Ethics Review Description:**

This paper looks like it's been done by AI.

**Ethics Review Flag:**

Yes

**Reviewer Confidence:**

3: The reviewer is confident but not certain that the evaluation is correct

**Scope:**

4: The work is relevant to the Web and to the track, and is of broad interest to the community

---

### Decision · Program_Chairs · 2024-01-22

**Decision:**

Accept (Oral)

**Comment:**

All reviewers agree that this paper, that uses attentional Hawkes processes to address the problem of comformity bias is an interesting and novel approach. Some concerns have been raised, which the authors mostly address in their comments. We hope they can incorporate the new material as required by the reviewers' feedback in the final version of the paper, if it gets accepted.